# Metformin as an Adjuvant to Photodynamic Therapy in Resistant Basal Cell Carcinoma Cells

**DOI:** 10.3390/cancers12030668

**Published:** 2020-03-13

**Authors:** Marta Mascaraque, Pablo Delgado-Wicke, Cristina Nuevo-Tapioles, Tamara Gracia-Cazaña, Edgar Abarca-Lachen, Salvador González, José M. Cuezva, Yolanda Gilaberte, Ángeles Juarranz

**Affiliations:** 1Departamento de Biología, Universidad Autónoma de Madrid, 28049 Madrid, Spain; martamascaraque@gmail.com (M.M.); pablo.delgado@uam.es (P.D.-W.); 2Instituto Ramón y Cajal de Investigaciones Sanitarias, IRYCIS, 28034 Madrid, Spain; 3Centro de Biología Molecular-Severo Ochoa (CBMSO/CSIC) and Centro de Investigación Biomédica en Red de Enfermedades Raras (CIBERER-ISCIII), Universidad Autónoma de Madrid, 28049 Madrid, Spain; cnuevo@cbm.csic.es (C.N.-T.); jmcuezva@cbm.csic.es (J.M.C.); 4Departmento de Dermatología, Hospital Barbastro, 22300 Huesca, Spain; tamara_gracia@hotmail.com; 5Facultad de Ciencias de la Salud, Universidad San Jorge, 50830 Villanueva de Gállego, Spain; edgarabarcalachen@gmail.com; 6Departmento de Medicina y Especialidades Médicas, Universidad de Alcalá, 28801 Madrid, Spain; salvagonrod@gmail.com; 7Servicio de Dermatología, Hospital Miguel Servet, 50009 Zaragoza, Spain; ygilaberte@gmail.com

**Keywords:** basal cell carcinoma, photodynamic therapy, resistance, metabolic markers, metformin

## Abstract

Photodynamic Therapy (PDT) with methyl-aminolevulinate (MAL-PDT) is being used for the treatment of Basal Cell Carcinoma (BCC), although resistant cells may appear. Normal differentiated cells depend primarily on mitochondrial oxidative phosphorylation (OXPHOS) to generate energy, but cancer cells switch this metabolism to aerobic glycolysis (Warburg effect), influencing the response to therapies. We have analyzed the expression of metabolic markers (β-F1-ATPase/GAPDH (glyceraldehyde-3-phosphate dehydrogenase) ratio, pyruvate kinase M2 (PKM2), oxygen consume ratio, and lactate extracellular production) in the resistance to PDT of mouse BCC cell lines (named ASZ and CSZ, heterozygous for *ptch1*). We have also evaluated the ability of metformin (Metf), an antidiabetic type II compound that acts through inhibition of the AMP-activated protein kinase (AMPK)/mammalian target of rapamycin (mTOR) pathway to sensitize resistant cells to PDT. The results obtained indicated that resistant cells showed an aerobic glycolysis metabolism. The treatment with Metf induced arrest in the G0/G1 phase and a reduction in the lactate extracellular production in all cell lines. The addition of Metf to MAL-PDT improved the cytotoxic effect on parental and resistant cells, which was not dependent on the PS protoporphyrin IX (PpIX) production. After Metf + MAL-PDT treatment, activation of pAMPK was detected, suppressing the mTOR pathway in most of the cells. Enhanced PDT-response with Metf was also observed in ASZ tumors. In conclusion, Metf increased the response to MAL-PDT in murine BCC cells resistant to PDT with aerobic glycolysis.

## 1. Introduction

Basal cell carcinoma (BCC) is the most common skin cancer worldwide, and its incidence rate has increased in recent decades [1]. BCC is generally slow-growing and rarely metastasizes; however, it can be highly destructive and mutilating to local tissues, and its recurrence rate at five years is quite high, about 20% [2,3]. 

The main treatment for BCC involves surgical modalities associated with disfigurement. Since BCC usually appears in photo-exposed areas, such as face or extremities, noninvasive therapies such as Photodynamic Therapy (PDT) have been developed for its treatment [3].

PDT is approved in the clinic for the treatment of several forms of non-melanoma skin cancer [4], and there are several clinical trials for gastrointestinal and prostate carcinomas, among other type of cancers [5,6]. PDT is a two-step therapy: (i) administration of a photosensitizer (PS) and (ii) tumor irradiation with light of a specific wavelength that generates reactive oxygen species (ROS) that destroy tumor cells [7,8,9,10]. One of the compounds approved for its use in oncological dermatology is methyl-aminolevulinate (MAL), a precursor of the endogenous PS protoporphyrin IX (PpIX). MAL is approved for the treatment of actinic keratosis (AKs) in the U.S. and the E.U. and for superficial and nodular BCC and Bowen’s disease in the E.U. [8,10].

However, PDT is not always effective and recurrences may occur after the treatment [3]. Normal differentiated cells depend primarily on mitochondrial oxidative phosphorylation (OXPHOS) to generate energy, but cancer cells switch this metabolism to an enhanced aerobic glycolysis, a process known as Warburg effect [11]. Changes during the reprogramming of cellular metabolism have been correlated, among other events, with changes in the expression of the β-catalytic subunit of the mitochondrial H^+^-ATP synthase (β-F1-ATPase), which is the bottleneck of OXPHOS, relative to the expression of glycolytic glyceraldehyde-3-phosphate dehydrogenase (GAPDH) in different neoplasia [12,13]. It has been described that Warburg effect could influence the response to antitumor drugs [14,15,16]. In addition, it has been observed (i) that pyruvate kinase M2 (PKM2) is highly expressed in cancers and promotes cell survival [17] and (ii) that phosphoinositide 3-kinase signaling via protein kinase B (AKT) is able to regulate glucose transporter expression (GLUT1), enhancing glucose capture [18]. In addition, the mammalian target of rapamycin (mTOR) seems to play a significant role in the metabolic reprogramming of tumor cells, specifically mTOR complex 1 (mTORC1), which is activated in the majority of cancers [19]. 

One possible solution to avoid recurrences in PDT is its combination with other treatments. Since changes in cell metabolism occur in cancer cells, the use of compounds that regulate the glucose metabolism could be an excellent option to reduce relapses. In this sense, metformin (Metf), used in type 2 diabetes mellitus, has become a promising drug in oncology, acting, particularly through inhibition of the AMP-activated protein kinase (AMPK)/mTOR pathway [20,21]. In skin cancer, it has been described that Metf inhibits tumor promotion (papillomas and squamous cells carcinomas) in overweight and obese mice downstream of mTORC1 [22]. In addition, other studies have demonstrated that Metf enhanced the cytotoxicity of PDT with 5-aminolevulinic acid in lung cancer cells [23] and reduced angiogenesis and tumor inflammation in combination with 20-tetra-sulfophenyl-porphyrin-PDT in Walker 256 carcinosarcomas induced in rats [24].

The present study evaluates the metabolic reprogramming in two BCC murine cell lines (named ASZ and CSZ) resistant to PDT and the adjuvant effect of Metf to this therapy in vitro on these two murine cell lines and in vivo in tumors generated by injecting ASZ cells. 

## 2. Results

### 2.1. Cells Survival Evaluation after PDT

The mouse cell lines named ASZ and CSZ were used, both heterozygous for the *ptch1* gene [25]. The cells, called parental (P), were subjected to 10 PDT cycles (1 mM methyl-aminolevulinate, followed by red light irradiation) to obtain resistant cells (10 G). Resistant cells were inoculated in immunosuppressed mice; the induced tumors were subcultured by explants, and a cell population called 10 GT was obtained (Appendix A) [2]. After generation of resistant populations of BCC murine cells (ASZ and CSZ), their resistance to PDT was validated, in terms of cell survival, by the 3-[4,5-dimethylthiazol-2-yl]-2,5- diphenyltetrazoliumbromide (MTT) assay. The obtained data confirmed, as expected, that 10 G populations of ASZ and CSZ cells were more resistant to PDT than their respective P populations. In addition, 10 GT CSZ cells were significantly more resistant than their respective P and 10 G populations; however, this was not observed with 10 GT of ASZ cells that showed a lower resistance than their corresponding P and 10 G (Figure 1a,b). For all the experiments, the corresponding controls were performed: untreated cells (cells without MAL or light irradiation) and cells treated with MAL (0.2 mM, 5 h) or red light alone (15.2 J/cm^2^); no cell toxicity was detected.

According to these results, we selected the 10 G population of ASZ and the 10 GT of CSZ cells as resistant cells to PDT to perform the rest of the experiments. In addition, to evaluate the synergic effect with Metf, conditions of MAL-PDT that induced in the P populations a DL30 (lethal dose of 30%) were selected (0.2 mM MAL and 7.6 J/cm^2^ in ASZ and 3.8 J/cm^2^ in CSZ cells). 

### 2.2. Proliferation Capacity and Metabolic Characterization

By using the clonogenic assay, we tested the proliferative capacity of each cell population by evaluating the size of the colonies formed: small (<1 mm), medium (1–2 mm), and large (>2 mm). The results obtained with ASZ were in agreement with those previously published by our group [2], indicating that P and 10 G of ASZ cells formed a higher number of small colonies than their respective CSZ cells. However, ASZ did not show differences in size between P and the resistant cells; the same happened with the colonies of CSZ. Therefore, we cannot associate an increase in cell proliferation with the resistance to PDT (Figure 2a).

We next evaluated the expression of the metabolic markers β-F1-ATPase, GAPDH, and PKM2 at the protein level by western blot (WB) (Figure 2b,c). The β-F1-ATPase/GAPDH ratio, a proteomic index of the overall mitochondrial metabolism of the cell, was significantly decreased in both resistant cell lines, suggesting a lower OXPHOS in these cells compared to their corresponding P cells (Figure 2b). In addition, the PKM2 (isoform of PK that promotes aerobic glycolysis) levels were significantly increased in 10 G of ASZ cells (Figure 2c) (Appendix A).

In order to determine oxygen consumption rate (OCR) due to the activity of the respiratory chain, oligomycin, 2,4-Dinitrophenol (DNP), rotenone, and antimycin were sequentially added to the cells (Figure 2d). After oligomycin application, OCR decreased (A) and maximal respiratory rate after DNP addition was achieved (B). Treatment with rotenone plus antimycin induced a strong decrease in OCR, setting the respiration near zero values (C). The results obtained indicated that P cells (of both cell lines) displayed higher OCR than their corresponding resistant populations (Figure 2d). In addition, we also determined the oligomycin sensitive respiration rate (OSR), which represents the activity of synthesis of ATP by complex V coupled to the respiratory chain. As expected, the resistant cells of both cell lines showed a decrease in OSR, being statistically significant in ASZ, suggesting a lower synthesis of ATP (Figure 2e). Finally, we measured the lactate extracellular production to complete the metabolism analysis. The 10 G of ASZ showed a higher lactate extracellular production than the P cells, according to the OXPHOS diminution observed. However, this was not appreciated in 10 GT of CSZ, where significant differences with P cells were not found (Figure 2f). A summary of these data is shown in Appendix A.

### 2.3. Metformin Treatment

The results shown above indicated that, in the PDT-resistant cells, the OXPHOS metabolism was lower than that of parental cells. In addition, the resistant ASZ population showed a higher glucose metabolism than that of parental cells. Therefore, we next evaluate the effects of Metf as potential treatment against BCC cells resistant to PDT. The results obtained indicated that Metf induced a significant decrease in cell survival in both P (ASZ and CSZ) cells with concentrations ≥ 75 µM (Figure 3a). The resistant cells were less sensitive to Metf. We selected this dose of 75 µM Metf to evaluate its effects in cell cycle progression by using flow cytometry. At 24 h after Metf treatment, a relevant arrest in G0/G1 associated with a decrease in S and G2 phase in all the cell lines was observed (Figure 3b). 

Furthermore, we also analyzed the changes in the metabolism of these cells after the Metf treatment. First, we evaluated the membrane potential of the cells by JC-1 staining after 24 h of Metf treatment (75 µM) (Figure 3c). When mitochondria are polarized, JC-1 is concentrated in the organelle, forming aggregates fluorescing in red. When mitochondria are depolarized, JC-1 cannot be concentrated in the mitochondria and is distributed in monomeric form, emitting a green fluorescence. The red/green fluorescence ratio was significantly increased in 10 GT of CSZ cells after 24 h of Metf treatment, indicating that Metf causes hyperpolarization of mitochondrial membrane in resistant cells by inhibition of the respiratory chain. The rest of the cells did not show substantial differences in the red/green fluorescence ratio between untreated and Metf treated cells. We also evaluated the β-F1-ATPase, GAPDH, and PKM2 protein expression in ASZ (Figure 3d) and CSZ (Figure 3e) cells by WB after Metf treatment. This treatment did not induce significant differences in the expression of such metabolic markers (β-F1-ATPase/GAPDH ratio and PKM2) between P and 10 G of ASZ cells. However, in 10 GT of CSZ, the β-F1-ATPase/GAPDH ratio was significantly increased after Metf treatment, suggesting a higher ATP production. This result was in concordance with the increase in the mitochondrial membrane polarization observed with JC-1 staining in 10 GT of CSZ cells treated with Metf. The levels of PKM2 after the treatment in CSZ were lower than in controls, being significantly decreased in the 10 GT cells (Appendix A).

The OCR after Metf treatment showed a significant increase in the maxima respiration (after addition of DNP (B)) in ASZ P cells, suggesting an intensification of OXPHOS. However, 10 G of ASZ cells displayed a total reduction of its OCR after the treatment. Both CSZ populations did not show significant differences in the maxima respiration (Figure 3f). This result was in concordance with that obtained in OSR after Metf treatment: ASZ P showed a small increase; 10 G of ASZ and P of CSZ cells displayed a significant decrease; and 10 GT of CSZ cells showed a small diminution (Figure 3g). Finally, all the cell lines showed a significant diminution in the lactate extracellular production, suggesting a reduction of the glycolytic flux after Metf treatment (Figure 3h). The set of results obtained hints at the 10 G of ASZ and P of CSZ cells stopped their metabolism (OXPHOS and glycolysis) after Metf treatment; the 10 GT of CSZ stopped the glycolytic flux and the P of ASZ cells increased the OXPHOS and stopped the glycolytic flux. To better clarify these results, a summary of all the data is shown in Appendix A.

### 2.4. Combined Treatment of Metf and MAL-PDT on BCC Cell Lines

We next explored whether the pretreatment with Metf improved the response to MAL-PDT of P and resistant PDT cells. We observed a significant decrease in cell survival after the combination of MAL-PDT (0.2 mM MAL and 7.6 J/cm^2^ in ASZ cell line and 3.8 J/cm^2^ in CSZ cell lines) and Metf (25–150 µM), which was also related to Metf concentration (Figure 4a,a’). The synergist/antagonism parameter DL (difference in logarithm) values (calculated as indicated in the Materials and Methods section) were larger than 2, confirming the synergistic effect of MAL-PDT and Metf (Figure 4b,b’). 

Since the combination of treatments improved the response to PDT, we evaluated if Metf treatment increased the intracellular PpIX production, since the amount of this heme metabolite is the basis of PDT action. PpIX content was determined by flow cytometry in P and resistant populations of both BCC cells prior incubation with MAL (basal level PpIX), after 24 h Metf treatment, after 5 h with 0.2 mM MAL, and after 24 h Metf treatment followed by 5-h incubation of 0.2 mM MAL (Figure 5a). Cells treated with Metf alone did not show differences in the production of PpIX with respect to the untreated control cells. All CSZ populations showed higher intracellular PpIX content after 5 h of MAL incubation with respect to untreated cells, being statistically significant. In the case of ASZ cells (P and 10 G), the production of PpIX, after MAL incubation, seemed to be evident, although this was not statistically significant with respect to untreated cells. The combined treatment (Metf + MAL) did not induce significant differences in PpIX production with respect to MAL alone, except for P of CSZ in which the combined treatment produced less PpIX than MAL alone.

In order to determine the mechanism by which Metf improved the response to PDT in the BCC cell lines, we analyzed the expression of the AMPK/mTOR pathway proteins by WB: control; 24 h after 75µM Metf; 24 h after PDT treatment (5-h incubation with MAL and 7.6 J/cm^2^ in ASZ and 3.8 J/cm^2^ in CSZ cells); and combination of Metf and MAL-PDT (Figure 5b). Combined treatment increased the pAMPK/AMPK ratio in all cell lines. In both ASZ populations (P and resistant), we also observed a decrease in the expression of pAKT and p-p70S6K, suggesting a modification in the AMPK/mTOR pathway. Changes in the expression of pAKT in CSZ P cells were not observed but that of p-p70S6K was reduced after the three treatments compared to control cells. In 10 GT of CSZ cells, the expression of pAKT and p-p70S6K was increased after PDT and after the combined treatments, suggesting that Metf does not inhibit the AMPK-mTOR pathway (Appendix A). 

### 2.5. Effect of Metf and MAL-PDT on Tumor Growth Inhibition In Vivo

Considering the obtained in vitro results, we selected the ASZ cell line to perform tumor studies because these cells showed better response to PDT with the addition of Metf than CSZ cell lines. P and 10 G of ASZ cells were inoculated in nude mice to induce tumor in order to evaluate the potential antitumor activity of Metf and MAL-PDT. The control group received drinking water, while treatment groups were exposed to Metf (200 µg/mL diluted in drinking water during the experiment starting at day 9) and to MAL-PDT (at day 15 of inoculation, 2 mM MAL was injected in 50 µL PBS (phosphate buffered saline), and after 4 h of dark incubation, mice were exposed to 25 J/cm^2^ of red light) alone or in combination (Metf + MAL-PDT). Figure 6a shows the results of the in vivo experiment where P cells were injected, and Figure 6b shows that with 10 G cells. In the tumors of P cells, the Metf group significantly inhibited tumor growth, compared to the untreated control P tumors. The antitumor effect of the combined treatment was slightly higher than that of Metf alone. The PDT treatment conditions used did not induce any antitumor effects, and the sizes of PDT treated tumors were similar to those of controls. In the tumors of 10 G, the tumors treated with PDT showed a significant increase after the treatment compared to that of untreated controls, confirming their resistance to PDT. However, the tumors exposed to the combined treatment showed a significantly lower size compared to these subjected to PDT alone, indicating that Metf contributed to the inhibition of tumor growth. The histological characteristics of tumors induced by P cells were the presence of fusiform cells formed by severe atypical keratinocytes with nuclear pleomorphism, even infiltrating adipose tissue and skeletal muscle fibers. This infiltrative phenotype has been also previously described for ASZ induced tumors in mice [26]. Similar results were obtained with the tumors induced by the 10 G of ASZ cells (Figure 6c). Both cell types induced tumors with “infiltrative-like” aspect that grew invasively beyond the dermis (Figure 6c). The treatment with Metf of the tumors formed by P cells provoked small areas of red blood cell extravasation but larger than those observed in the 10 G tumors. These areas were also observed in the tumors treated with PDT, being larger in those induced by P than in 10 G cells. The combined treatment (Metf + PDT) induced extensive areas of blood extravasation in both types of tumors. In addition, extensive areas of cell death observed, both by hematoxylin and eosin (H&E) staining and by the additional TUNEL (terminal deoxynucleotide transferase mediated X-dUTP nick end labeling) assay (fluorescing in green), were detected in the treated tumors, particularly in those subjected to the combined treatment (Figure 6c). 

## 3. Discussion

PDT is an efficient treatment for BCC; however relapses occur in 24% of cases 5 years after treatment [27]. Therefore, understanding the resistant mechanism to PDT constitutes a very important goal for the management of BCC. It is known that cancer cells rewire their metabolism to promote growth and survival by increasing glucose consumption rate and lactate production even in the presence of oxygen and with fully functioning mitochondria. This process is known as Warburg effect [18,28]. In addition, during the last years, many articles have appeared indicating the Warburg effect may be potentially related with drug resistance treatment in cancer [14,29,30]. Therefore, in this study, we have evaluated the potential metabolic changes that occur in BCC cells resistant to PDT, associating the resistance with a glycolytic metabolism. The BCC cells used, ASZ and CSZ, were isolated from tumors induced in mice with different genetic background but all heterozygous for the *ptch* gene implicated in the development of most BCCs (ASZ isolated from *ptch1+/* and CSZ isolated from *ptch1+/-,* K5-CrePR, and p53fl/fl) [25]. These cells, called parental (P), were subjected to 10 MAL-PDT cycles to obtain resistant cell populations (10 G). Resistant cells were inoculated in immunosuppressed mice, and the induced tumors were subcultured by explants to obtain a cell population called 10 GT [2]; this process was carried under the basis that the in vivo microenvironment promotes the selection of cells with higher tumorigenic properties. We confirmed the resistance of the different populations to PDT, and the results obtained indicated that surprisingly 10 GT of ASZ cell line was more sensitive to MAL-PDT than their corresponding P cells. These results indicated that, in this case, the reselection in the tumor environment did not favor the resistant cells, as opposed with that described in the bibliography with other cell lines resistant to other chemotherapeutic compounds [31].

We have observed significant differences in the expression of the metabolic markers studied between resistant and P cells, including a lower ratio of β-F1-ATPase/GAPDH, that indicates loss of bioenergetic activity of the mitochondria in both resistant cell lines [15]; increase in PKM2 expression in 10 G of ASZ cells, a regulator of aerobic glycolysis [32]; OCR and OSR diminution, indicating less OXPHOS activity; and increase in the lactate extracellular production in 10 G of ASZ, suggesting a higher glycolytic flux [33]. These facts evidenced changes in the metabolism of resistant cells towards an aerobic glycolysis, being in agreement with other publications supporting the fact that the metabolic markers predict the response to different cancer treatments [15,29,30]. However, although it has been described that the aerobic glycolytic metabolism is an advantage for cancer cell proliferation [34], here, we have not found differences in the proliferation rate between P and PDT-resistant cells. 

One regulator of the Warburg effect is mTOR, which plays a central role in the regulation of cellular energy homeostasis [19,35,36]. Metf is a widely used drug for the treatment of type 2 diabetes that inactivates mTOR, with demonstrated antitumoral properties [21,22,37]. Our obtained data indicated that doses of Metf at micromolar concentrations do not substantially affect the survival of the ASZ and CSZ cells but induced arrest in the proliferation (decrease of S-G2/M and increased G0/G1 phases). This effect has been described in other tumor cells including breast cancer and glioma human cell lines [38,39]. In addition, it is known that Metf inhibits the electron transport chain [40]. Here, we showed that the treatment with Metf of 10 G of ASZ and P of CSZ cells stopped their OXPHOS and glycolytic metabolism due to a reduction in their OSR and lactate extracellular production. In 10 GT of CSZ, Metf produced a blockage of the glycolytic flux due to a diminution in PKM2 level and lactate extracellular production and to a small inhibition of the OXPHOS observed by the hyperpolarization of the mitochondria. Finally, in the P of ASZ, the OXPHOS (show a higher maximal respiration and small increase in its OSR) increased and the glycolytic flux stopped due to the reduction of lactate extracellular production. Therefore, in the BCC cells used in this work, Metf plays a relevant role in the inhibition of the glycolytic flux and in the inhibition of OXPHOS in 10 G of ASZ and P of CSZ cells. 

We have tested if the pretreatment with Metf improved the response to MAL-PDT in the BCC cells and particularly in the PDT-resistant cells. The treatment with Metf followed by MAL-PDT induced a high toxicity in all the cells, including those resistant to PDT. This major cytotoxicity was due to a synergic effect of Metf and MAL-PDT and not to an additive effect. Our results are in agreement with other studies and indicate that Metf improves the cytotoxicity induced by PDT with 5-aminolevulinc acid in human oral squamous cell carcinoma [41] or by chemotherapy using paclitaxel or cisplatin in other cell lines [42,43].

MAL-PDT is related to the production and accumulation of PpIX in the tumoral cells [10]. However, Metf did not increase the accumulation of PpIX into cells, so this is not the mechanism by which the combination of these treatments increases its cytotoxicity. Even, in P of CSZ cells, a significant diminution in the production of PpIX was observed when the cells were pretreated with Metf. It is tempting to consider that the modulation of the mTOR pathway and OXPHOS by Metf (which also occurs in mitochondria where PpIX is produced) is implicated.

To characterize the mechanism of action by which the pretreatment with Metf improved the response to PDT, we analyzed the AMPK-mTOR pathway, since it is a very important target of the mechanism of action of Metf [21,40]. The pAMPK levels after Metf treatment were higher than those of controls in all cell lines, although the ratio pAMPK/AMPK was not significantly increased. However, pretreating 24 h with Metf before PDT increased the pAMPK/AMPK ratio in all cell lines. Several studies relate the Metf treatment with an activation of AMPK and the modulation of downstream effectors implicated in cellular growth and metabolism [21,22,38]. In both ASZ cells, the increase of pAMPK after the combined treatment was associated with a decrease of pAKT/AKT ratio and p-p70S6K/p70S6K ratio, indicating a suppression of the mTOR pathway. Other studies that combine Metf with chemotherapeutic agents, such as paclitaxel or 5-FU, indicate a suppression of the mTOR pathway [44,45]. However, 10 GT of CSZ, after this combined treatment increased the expression of pAKT/AKT ratio and p-p70S6K/p70S6K ratio, indicating an over activation of the mTOR pathway. Therefore, the improvement of the response to PDT after Metf treatment in these cell lines could be related to the hyperpolarization of the mitochondria.

Considering the obtained in vitro results, we selected the ASZ cell line to perform tumors studies because these cells showed better response to PDT with the addition of Metf than CSZ cell lines. The treatment with PDT did not induce a significant decrease in tumor volume relative to controls. Although treatment with Metf decreases tumor growth, as it has been previously described [41,46,47], we did not show any significant differences in the tumor volume with the combined treatment under the treatment conditions applied. However, the tumors induced by 10 G ASZ cells undergoing combined treatment showed a reduction in their growth rate in relation to those treated with PDT alone, probing that Metf modulates the exponential growth of these cells after PDT.

In summary, the present study shows that MAL-PDT-resistant BCC cells exhibited an aerobic glycolysis, which seems to be related to their lack of response to PDT. The pretreatment with Metf seems to have a synergic effect with MAL-PDT against BCC cells. 

## 4. Materials and Methods 

### 4.1. Cell Culture

The cell lines used were obtained from BCCs induced by UV irradiation in a ptch1^+/-^ mouse (ASZ001, ASZ) and from spontaneous tumor developed in a ptch1^+/-^, K5-CrePR, and p53fl/fl mouse (CSZ1, CSZ) [25]. These cells, called parental (P), in a previous work were subjected to 10 MAL-PDT cycles to obtain resistant cells (10 G), and 10 G cells were inoculated in immunosuppressed mice; the induced tumors were subcultured by explants, and a cell population called 10 GT was obtained (Appendix A) [2]. Cells were grown in DMEM (Dulbecco’s modified Eagle’s medium high glucose) supplemented with 10% (*v/v*) fetal bovine serum (FBS) and 1% antibiotic (penicillin, 100 units/mL; streptomycin 100 mg/mL), all from Thermo Fisher Scientific Inc (Rockford, IL, USA). Cell cultures were performed under standard conditions of 5% CO2, 95% humidity, and 37 °C and propagated by treatment with 1 mM EDTA/0.25% Trypsin (*w/v*).

### 4.2. Treatments

For PDT, Methyl-aminolevulinate (MAL) (Sigma-Aldrich, St. Louis, MO, USA) was prepared at an initial concentration of 10 mM in deionized sterile water. For phototreatments, when cells reached a 60–70% confluence, they were incubated with 0.2 mM MAL in DMEM culture medium without FBS for 5 h. Afterwards, cells were irradiated for variable light doses (1.5 to 15 J/cm^2^) by using a red-light emitting diode source (WP7143 SURC/E Kingsbright, Angels, CA, USA) with an irradiation intensity of 6.2 mW/cm^2^ and an emission peak at λ = 634 ± 20 nm. To minimize light refraction, cells were irradiated from the bottom of the plates. After irradiation, cells were incubated in DMEM with 10% FBS for 24 h until evaluation.

Stock solution of metformin (Metf) (European Pharmacopoeia, Sigma-Aldrich, St. Louis, MO, USA) (6 mM) was prepared in dimethyl sulfoxide (DMSO) (Panreac, Barcelona, Spain), and the work solution was obtained in DMEM with 10% (*v/v*) FBS. The final concentration of DMSO was always lower than 0.5% (*v/v*). The cells were treated for 24 h with Metf (25–150 µM).

Combined treatments were carried out when cells reached a 50–60% confluence. First, the cells were incubated for 24 h with Metf. Afterwards, the medium with Metf was replaced by that of MAL (0.2 mM) and further incubated for 5 h. Subsequently, cells were irradiated with the red light source and the medium with MAL was replaced by fresh DMEM with 10% FBS and incubated for 24 h until evaluation [48]. 

### 4.3. Cellular Toxicity

Cell viability was determined using the MTT (3-[4,5-dimethylthiazol-2-yl]-2,5- diphenyltetrazoliumbromide) assay (Sigma-Aldrich, St. Louis, MO, USA). MTT solution (100 μg/mL) was added to cell cultures and incubated at 37 °C for 3 h. After incubation, the formazan precipitate was dissolved with DMSO and optical density was measured in a SpectraFluor (Tecan, Bradenton, FL, USA) plate reader at 542 nm. Cellular toxicity was expressed as the percentage of surviving cells relative to that of the non-treated (control) cells.

### 4.4. Cell Proliferation

Proliferation rate was determined by the clonogenic assay. Cells were seed at 50 cell/mL per well in P6 plates, and they grew for 7 days. Then, the cells were fixed and stained with 0.2% crystal violet (Sigma-Aldrich, St. Louis, MO, USA) in 2% ethanol in distilled water for 20 min under constant shaking at room temperature. Finally, the plates were washed with PBS (phosphate buffered saline), and air dryer and colonies were counted and classified in groups according to their size as: small (<1 mm); medium (1–2 mm); and large (>2 mm).

### 4.5. Western Blots

For western blots (WB), cellular extracts were obtained with RIPA buffer with Triton, pH 7.4 (Bioworld), containing phosphatase (PhosSTOP EASYpack, Roche, Mannheim, Germany) and protease (complete ULTRA tablets Mini EDTA-free EASYpack, Roche) inhibitors, following the manufacturer’s instructions. Protein concentration was determined by BCA Protein Assay Kit (Thermo Scientific Pierce, Rockford, IL, USA). Cellular extracts (30 µg protein/lane) were diluted in Laemmli buffer (Bio-Rad, Hercules, CA, USA) and heated for 5 min at 98 °C. Electrophoresis was performed using acrylamide/bisacrylamide gels in denaturing conditions (SDS-PAGE) and transferred to polyvinylidene difluoride (PVDF) membranes (Bio-Rad), performed using a Transblot Turbo system (Bio-Rad). Membranes were blocked in skimmed milk in 0.1% TBS-Tween 20 and then incubated with antibodies against pAMPK, (Thr172), AMPK, pAKT (ser473), AKT, phosho-p70S6K (Thr 389) and p70S6K (Cell Signaling Tecnology, Inc, Danvers, MA, USA), GAPDH, PKM2, α-tubulin (Abcam, Cambridge, UK), and β1-ATPase (clone 17/9–15G1) (Dr. Cuezva) and then further washed and incubated with the corresponding peroxidase-conjugated secondary antibodies (Thermo Fisher, Rockford, IL, USA). Protein bands were visualized by chemiluminiscence (ECL Plus Kit, Amersham, Little Chalfont, UK) using the high-resolution ChemiDocTR XRS+ system (Bio-Rad) and digitalized using Image Lab version 3.0.1 software (Bio-Rad).

### 4.6. Determination of Cellular Respiration and Rates of Glycolysis

Oxygen consumption rate (OCR) was determined by using XF24 Seahorse flux analyzer. For that, cells were cultured in an XF 24-well cell culture microplate (Agilent technologies, Santa Clara, CA, USA) at a density of 30 × 10^3^ cells/well in 200 µL of growth medium for 24 h. Then, 700 µL of XF Base Medium (Agilent technologies) supplemented with 25 mM glucose (Sigma-Aldrich, St. Louis, MO, USA), 1 mM pyruvate (Sigma), and 2 mM glutamine (Life Technologies, Carlsbad, CA, USA) were added, and the plate was incubated for 1 h at 37 °C without CO_2_. The final concentration and order of injected substances was 6 µM oligomycin (inhibits ATP synthase), 0.75 mM 2,4-Dinitrophenol (DNP) (an uncoupling agent that collapse the proton gradient and disrupts the mitochondrial membrane potential), 1 µM rotenone (inhibits complex I of the electron transport chain), and 1 µM antimycin (inhibits complex III of the electron transport chain) (all from Sigma-Aldrich, St. Louis, MO, USA). To determine the rates of glycolysis, the initial rates of lactate production were determined by the enzymatic quantification of lactate concentrations in the culture medium. Culture medium was replaced by fresh medium supplemented with 1% FBS 1 h before the measurement; 200 μL of culture medium samples was taken at different intervals (60 and 120 min) and precipitated with 800 μL of cold perchloric acid, incubated on ice for 1 h, and then centrifuged for 5 min at 11,000 × g at 4 °C to obtain a protein-free supernatant. The supernatants were neutralized with 20% (*w/v*) KOH and centrifuged at 11,000 × g and 4 °C for 5 min to sediment the KClO_4_ salt. Lactate levels were determined spectrophotometrically by following the reduction of NAD^+^ at A340 after the addition of 10 µL of lactate dehydrogenase (Sigma-Aldrich, St. Louis, MO, USA).

### 4.7. Detection of JC-1 Fluorescence 

JC-1 is a sensitive marker for mitochondrial membrane potential that forms aggregates in healthy mitochondria fluorescing in red; when the membrane potential decreases, JC-1 becomes monomers showing a green fluorescence. Therefore, the ratio of red/green fluorescence indicates the state of the mitochondrial potential [49]. JC-1 (5,5′6,6′- tetrachloro- 1,1′,3,3′- tetraethyl- benzamidazole carbo-cyanine iodide) (Invitrogen, ThermoFisher Scientific, Waltham, MA, USA) was prepared at 1 mg/mL in DMSO as a stock solution, aliquoted, and stored at −20 °C. For each analysis, cells were grown on coverslips and, 24 h after, Metf treatment cells were incubated with 1 µM JC-1 for 15 min at 37 °C. Then, cells were briefly washed with PBS, mounted on slides, and observed in situ with fluorescence using blue and green excitation light.

### 4.8. Cell Cycle

Cell distribution throughout the cell cycle phases was studied by flow cytometry (Cytomics FC500, 1 laser, Beckman Coulter, Corston, UK). Cells at 24 h after Metf treatments were trypsinized, fixed in 70% ethanol, and washed with PBS by centrifugation. The pellet was resuspended in 50 μL of DNAprep kit (Beckman Coulter) and in 1 mL of propidium iodide and incubated for 30 min at 37 °C.

### 4.9. Production of PpIX

The production of PpIX was evaluated by flow cytometry after cell incubation with 0.2 mM MAL for 5 h. Afterwards, cells were trypsinized, centrifuged 7 min at 1800 rpm, and fixed with 1% formaldehyde in PBS at room temperature for 10 min. Fixing solution was removed, washing the cells twice with PBS by centrifugation; resuspended in clean PBS; and kept in the dark at 4 °C until evaluation. PpIX emission measurements were obtained employing the flow cytometer FC500 Cytomics (λ_exc_ = 625 nm; λ_em_ = 670 nm). Fluorescence intensity was determined for 10^4^ cells per cellular population.

### 4.10. Evaluation of the Synergistic Effect after Combined Treatment

A statistical model was used to determine the magnitude of synergy observed after the combined treatment in both cell lines [50]. The model is based on the assumption that Metf and MAL-PDT have distinct mechanisms of action. In such a system, the additive effect of two treatments is the product of the cell survival percentage (CSP) of each treatment: CSP_add_ = CSP_Metf_ X CSP_PDT_. The calculated CSP_add_ was then compared to the observed CSP_comb_ by the synergy/antagonism parameter difference in logarithms (DL) between the observed CSP_comb_ and the calculated CSP_add._

DL = (log CSP_Metf_ + log CSP_PDT_) – log CSP_comb_

A DL < 2 indicates a synergistic effect, whereas an additive effect occurs when DL is 2.

### 4.11. In Vivo Experiments

For the in vivo assays, 6-week-old athymic nude-Foxn1nu mice (Envigo, France, Barcelona, Spain) were used. Mice were classified randomly in 2 groups with 6 mice per group. One group was inoculated in both flanks with 1.5 x 10^6^ P of ASZ cells in 50 µL of PBS and 50 µL of Matrigel (Corning, Bedford, MA, USA), and the other group was inoculated in both flanks with the same number of 10 G of ASZ cells [2]. All cells formed tumors of approximately 30 mm^3^ in 7 days after cell inoculation. In each group, mice were randomly distributed into two subgroups (3 mice per subgroup) and treated or not with 200 µg/mL Metf (diluted in drinking water and administered throughout the experiment starting at day 9 after inoculation). The right tumor of each mouse was treated with PDT at day 15 after inoculation when the tumor volumes reached a size of 100–150 mm^3^. For PDT, 50 µL PBS containing 2 mM MAL was administered by intratumoral injection, and after 4 h of dark incubation, tumors were exposed to 25 J/cm^2^ of red light. During the subsequent days, the animals were monitored measuring the progressive increase of tumor size with an automatic caliper. To calculate the tumor volume, all the lobules formed from each inoculation were considered and determined as follows: V = π/6 × (length × width × depth).

When the tumor reached the maximum established volume of 500 mm^3^, mice were sacrificed with CO_2_. Tumors were surgically removed and fixed with 3.7% formaldehyde (Panreac, Barcelona, Spain) in PBS, included in paraffin, sectioned, and stained with hematoxylin and eosin (H&E) for pathological analysis.

Apoptosis was determined by using TUNEL (terminal deoxynucleotide transferase mediated X-dUTP nick end labeling) assay. Briefly, tissue sections (5 µm) were placed on silanized slides. The sections were deparaffinized and rehydrated and then treated with Proteinase K (20 µg mL^−1^, 15 min, room temperature) followed by washing 2 times with PBS. Afterwards, the sections were incubated for 1 h at 37 °C with the in situ Cell Death Detection Kit (Roche Diagnostics GmbH, Roche Applied Science, Mannheim, Germany), according to the manufacturer’s instructions. For nuclear counterstaining, the samples were incubated with Höechst-33258 (Sigma, Darmstadt, Germany) at a final concentration of 1 µg/mL and mounted with Vectashield mounting medium (Vector, Burlingame, CA, USA).

All the experimental procedures with animals were carried out in compliance with the guidelines in RD 53/2013 (Spain) and was approved by the Ethics Committee from Consejo Superior de Investigaciones Científicas (CSIC, Madrid, Spain) and Comunidad Autónoma of Madrid (CAM, Consejería de Medio Ambiente; Register number: ES80790000188) in the frame of the projects FIS-PI15/00974 and FIS-PI18/00708 supported by the Spanish Ministerio de Economía y Competitividad.

### 4.12. Optical Microscopy

Microscopic observations were carried out using an Olympus BX61 epifluorescence microscope equipped with filter sets for fluorescence microscopy: ultraviolet (exciting filter BP360–390), blue (exciting filter BP460–490), and green (exciting filter BP510–550). Photographs were obtained with the digital camera Olympus CCD DP70 and processed using the Adobe Photoshop CS5 extended version 12.0 software (Adobe Systems Inc., San Jose, CA, USA).

### 4.13. Statistical Analysis

Data were expressed as the mean value of at least three experiments ± standard deviations (SD). The statistical analysis was carried out with the version 6.05 of the program GraphPad Prism (GraphPad Software Inc, USA) used to make graphical representations and the statistic. The statistical differences were determined using in general, analysis of variance (ANOVA, Chicago, IL, USA) and post hoc Bonferroni’s test (*p* < 0.05). For tumor volume, nonparametric Kruskall–Wallis test was used and *p* < 0.05 was considered statistically significant. 

## 5. Conclusions

The MAL-PDT resistant murine BCC cells to exhibit a glycolytic metabolism, which seems to be related to their lack of response to PDT. The pretreatment with Metf shows a synergic behavior with MAL-PDT against BCC cells and induces the activation of the AMPK, suppressing the mTOR pathway. The combined treatments (Metf + MAL-PDT) in vivo in the PDT-resistant tumors reduces the exponential growth of the tumors observed after PDT alone.

## Figures and Tables

**Figure 1 cancers-12-00668-f001:**
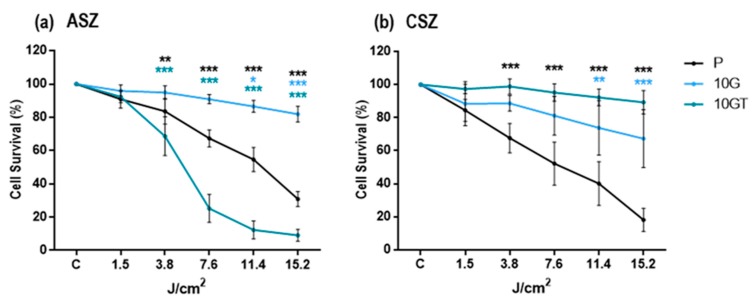
Cell survival after Photodynamic Therapy (PDT): Survival of P, 10 G, and 10 GT populations of (**a**) ASZ and (**b**) CSZ cell lines subjected to methyl-aminolevulinate (MAL)-PDT and evaluated by the 3-[4,5-dimethylthiazol-2-yl]-2,5- diphenyltetrazoliumbromide (MTT assay). MTT test was performed 24 h after PDT treatment (0.2 mM MAL for 5 h and subsequently exposed to variable doses of red light). The 10 G population showed the highest resistance to treatment in ASZ cell lines, whereas in CSZ, it was the 10 GT population. Values were represented as mean ± SD (* *p* < 0.05; ** *p* < 0.01; *** *p* < 0.001) (*n* = 5).

**Figure 2 cancers-12-00668-f002:**
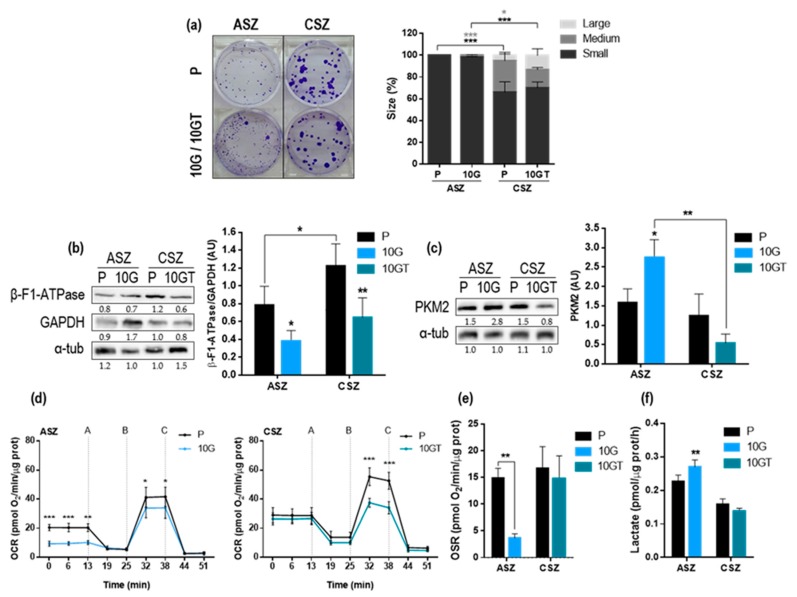
Proliferation capacity and metabolic characterization of Basal Cell Carcinoma (BCC) cells: (**a**) For the clonogenic assay, 50 cells/mL were seeded in each plate of 6 wells, and 7 days later, the colonies formed were stained with 0.2% crystal violet. Colonies were classified in relation to their diameter: small (<1 mm), medium (1–2 mm), and large (>2 mm) (*n* = 3). (**b**) Expression of the metabolic markers β-F1-ATPase and GAPDH (glyceraldehyde-3-phosphate dehydrogenase) analyzed by western blot (WB); alphatubulin was used as loading control; and the ratio of β-F1-ATPase/GAPDH indicates the use of glucose by the cells, which was significantly lower in the resistant comparing to that of P cells (*n* = 5). (**c**) Pyruvate kinase M2 (PKM2) levels were higher in 10 G of ASZ compared to the P cells (*n* = 3). (**d**) Oxygen consumption rate (OCR) measurements over time (min) were determined by using an extracellular flux analyzer after the sequential addition of oligomycin (A), 2,4-Dinitrophenol (DNP) (B), and rotenone + antimycin (C) (*n* = 4). (**e**) Oligomycin-sensitive respiration, which represents the activity of oxidative phosphorylation (OXPHOS), was calculated as basal respiration – oligomycin respiration (*n* = 4). (**f**) Rates of lactate production determined spectrophotometrically (*n* = 6). Values were represented as mean ± SD (* *p* < 0.05; ** *p* < 0.01; *** *p* < 0.001).

**Figure 3 cancers-12-00668-f003:**
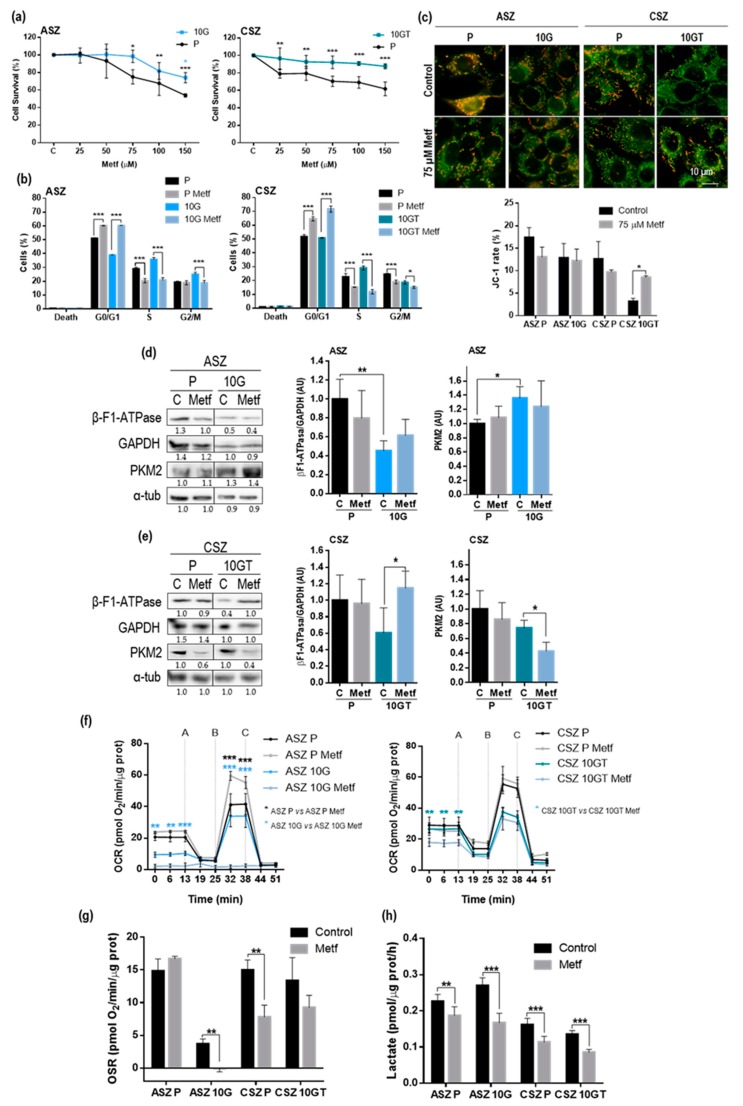
Metformin treatment: (**a**) Cell survival 24 h after Metf treatment in P and resistant populations of BCC cells evaluated by the MTT test. The P cells were dose dependent on Metf, and the resistant were less sensible than P cells. (**b**) Effect of 24 h treatment of Metf on cell cycle progression in P and resistant populations of ASZ and CSZ cells: Cell cycle distribution was analyzed by flow cytometry. Metf treatment induced a significant increase in the G1-G0 and a decrease in the S phases of all cell populations. (**c**) Mitochondrial membrane potential determined by JC-1 ratio (J-aggregate fluorescence/J-monomer fluorescence): The green and red fluorescence indicate J-monomers (low mitochondrial membrane potential) and J-aggregate (high mitochondrial membrane potential), respectively (*n* = 3). (**d**–**e**) Expression of the glycolytic markers (β-F1-ATPase/GAPDH ratio and PKM2) analyzed by WB in ASZ (Figure 3d) and CSZ (Figure 3e) cells (*n* = 5); alfa tubulin was used as loading control. (**f**) Real-time analysis of OCR in BCC cells after 24 h with 75 µM Metf and the sequential addition of oligomycin (A), 2,4-dinitrophenol (DNP) (B) and rotenone with antimycin (C) to the cells (*n* = 4). (**g**) Oligomycin sensitive respiration (OSR) after 24 h with 75 µM Metf, which represents the activity of OXPHOS, was calculated as basal respiration – oligomycin respiration (*n* = 4). (**h**) Rates of lactate production determined spectrophotometrically after 24 h with 75 µM Metf (*n* = 6). Values were represented as mean ± SD (* *p* < 0.05; ** *p* < 0.01; *** *p* < 0.001).

**Figure 4 cancers-12-00668-f004:**
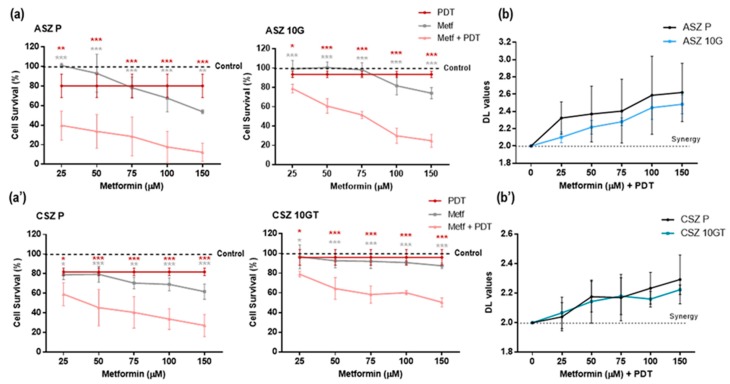
Combined treatment of Metf and MAL-PDT on cell viability: Cells were treated for 24 h with Metf (25–150 µM) and then subjected to MAL-PDT (0.2 mM MAL and 7.6 J/cm^2^ in ASZ cells and 3.8 J/cm^2^ in CSZ cells). Cell survival was evaluated by the MTT test. (**a**) The results obtained showed a decrease in the cell survival after the combined treatment compared to that obtained after Metf or PDT alone in (**a**) ASZ and (**a’**) CSZ cell lines. (**b**) Combined treatment provided a synergistic effect on cell viability in (**b**) ASZ and (**b’**) CSZ cell lines. The synergy/antagonism parameter DL (difference in logarithm) was calculated as follows: DL = (log cell survival percentage Metf + log cell survival percentage PDT) – log cell survival percentage combination. Values were represented as mean ± SD (* *p* < 0.05; ** *p* < 0.01; *** *p* < 0.001) (*n* = 5).

**Figure 5 cancers-12-00668-f005:**
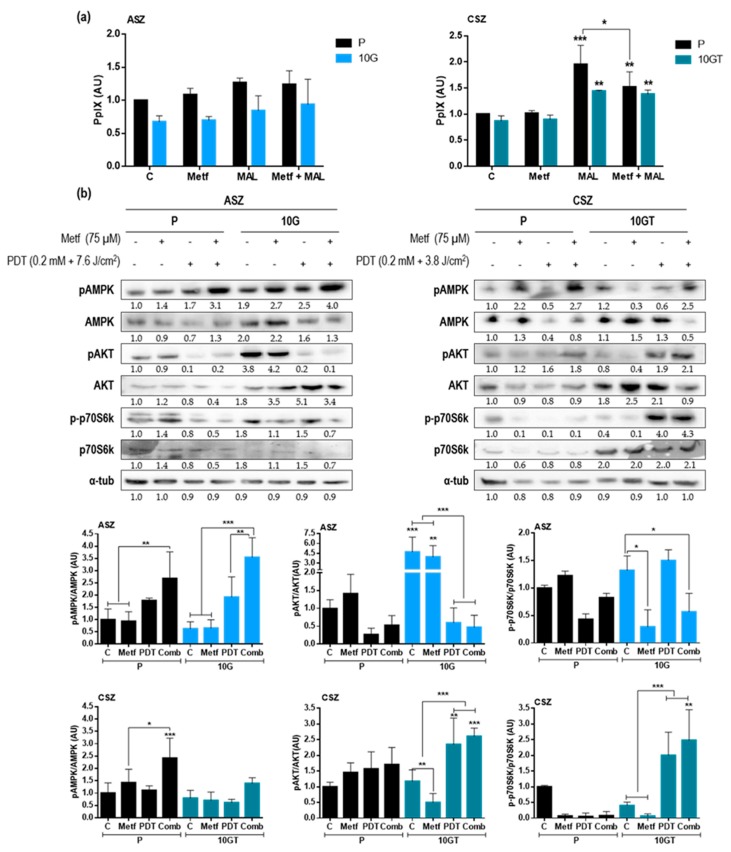
Effect of the combined treatment of Metf and MAL-PDT on BCC cell lines: (**a**) PpIX production was evaluated by flow cytometry after incubation with Metf (24 h, 75 µM), MAL (5 h, 0.2 mM), and Metf and MAL (24 h followed by 5 h, respectively). (**b**) Expression by western blot of AMP-activated protein kinase (AMPK)-mammalian target of rapamycin (mTOR) pathway components: pAMPK, AMPK, pAKT, AKT, p-p70S6K and p70S6K after treatments (Metf, MAL-PDT, and Metf plus MAL-PDT). A representative experiment of each marker is shown, and pAMPK/AMPK, pAKT/AKT, and p-p70S6K/p70S6K densitometry graphics of both P and resistant populations of ASZ and CSZ cells are shown. Alfa tubulin was used as loading control. For each cell population, 4 conditions were evaluated: control; 24 h after 75µM Metf; 24 h after PDT treatment (5 h incubation with MAL and 7.6 J/cm^2^ in ASZ and 3.8 J/cm^2^ in CSZ cells); and combination of Metf and MAL-PDT. Values were represented as mean ± SD (* *p* < 0.05; ** *p* < 0.01; *** *p* < 0.001).

**Figure 6 cancers-12-00668-f006:**
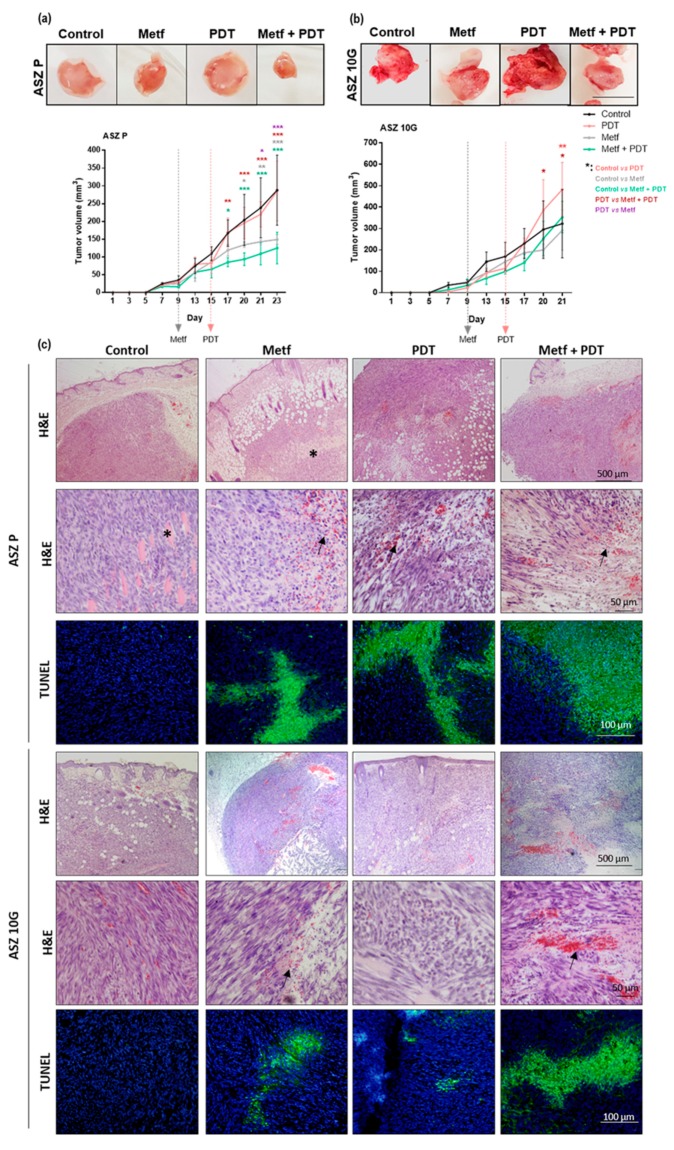
Effect of the treatments with Metf and/or PDT of ASZ tumors induced in mice: Photographs of the tumors at the time of sacrifice and the evolution of tumor volumes over time after the treatments in (**a**) P and (**b**) 10 G ASZ tumors. At day 9, when the tumors reached a volume of 50 mm^3^, they were treated with Metf (200 µg/mL diluted in drinking water along the rest of the experiment). At day 17, when the untreated group reached a volume of 100–200 mm^3^, the tumors were subjected to PDT or to Metf-PDT (2 mM MAL injected in 50 µL PBS, 4 h of incubation, and 25 J/cm^2^ of red light). Tumor volume was measured every two days with a caliper. Values were represented as mean ± SD (* *p* < 0.05; ** *p* < 0.01; *** *p* < 0.001) (*n* = 3). Scale bar = 10 mm. (**c**) Representative photographs of tumor sections at the end of the experiments stained with hematoxylin and eosin (H&E) (low and high magnification) and stained with the TUNEL (terminal deoxynucleotide transferase mediated X-dUTP nick end labeling) assay. The H&E showed that the tumors were formed by atypical keratinocytes infiltrating skeletal muscle fibers (asterisk). The treatment with Metf, PDT, and especially Metf + PDT provoked an increment of red blood cell extravasation in the dermis (black arrows) and extensive areas of cell death. The cell death areas were better observed after the TUNEL staining; dead cells appeared fluorescing in green particularly after the combined treatments applied in P tumors. Nuclei were counterstained with Höechst fluorochrome and fluoresced in blue.

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
