# Peer review of "Metformin as an Adjuvant to Photodynamic Therapy in Resistant Basal Cell Carcinoma Cells"

_cancers, 2020, doi:10.3390/cancers12030668_

Round 1
Reviewer 1 Report
The manuscript “Metformin as an adjuvant to photodynamic therapy in resistant basal cell carcinoma cells” describes the use of Metformin and methyl-aminolevulinate to treat Basal cell Carcinoma, the study is well organized, the results properly analyzed and the conclusion are appropriate, considering the data obtained.
The use of Metf as adjuvant to PDT is not new and some references to this matter should be added, for example: (J Lasers Med Sci. 2019;10(3):241-250. doi:10.15171/jlms.2019.39) (Journal of Photochemistry and Photobiology B: Biology 138 (2014) 80–91)or. (J Lasers Med Sci 2019 Summer;10(3):241-250 )
Author Response
- Referee 1
The manuscript “Metformin as an adjuvant to photodynamic therapy in resistant basal cell carcinoma cells” describes the use of Metformin and methyl-aminolevulinate to treat Basal cell Carcinoma, the study is well organized, the results properly analyzed and the conclusion are appropriate, considering the data obtained.
- The use of Metf as adjuvant to PDT is not new and some references to this matter should be added, for example: (J Lasers Med Sci. 2019;10(3):241-250. doi:10.15171/jlms.2019.39) (Journal of Photochemistry and Photobiology B: Biology 138 (2014) 80–91)or. (J Lasers Med Sci 2019 Summer;10(3):241-250 ).
According to the suggestion made by the referee, we have added an additional paragraph in the introduction in relation with the articles related to Metformin and PDT (line 71) as follows: “In addition, other studies have demonstrated that Metf enhanced the cytotoxicity of PDT with 5-aminolevulinic in lung cancer cells [23] and also reduced angiogenesis and tumor inflammation in combination with 20-tetra-sulfophenyl-porphyrin-PDT in Walker 256 carcinosarcomas induced in rats [24].”
Reviewer 2 Report
The authors presented a very interesting study regarding the combined usage of PDT and the drug Metformin, tipically used to lower the blood sugar in type 2 diabetes. Metformin is in fact a first‐line antihyperglycaemic not commonly used to treat cancer, especially BCC tumor type, so this represent a novelty in the field.
In their previous paper the authors developed a very smart strategy to induce PDT resistant cell lines (“Characterisation of resistance mechanisms developed by basal cell carcinoma cells in response to repeated cycles of Photodynamic Therapy” Sci Rep. 2019; 9: 4835.) and in this second article they use their model to prove the possible combination of PDT and Metf to increase cell death in case of tumor resistance.
Nevertheless, in the study several hypotheses based on previous and related literature, have not been proven probably due to the nature of the murine cell lines selected, their variability and the lack of a positive control such as resistant patient derived tumor cells.
For instance, the authors evaluated the potential metabolic changes that occur in BCC cells resistant to PDT, associating the resistance with a glycolytic metabolism, in order to get a rationale for the usage of an antihyperglycaemic drug. The results show changes in the metabolism of resistant cells towards an aerobic glycolysis, neverthless although it has been described that the aerobic glycolytic metabolism is an advantage for the cancer cell proliferation, the authors found no differences in the proliferation rate between parental and PDT resistant cells.
In addition, the hypothesis that Metf works by inhibiting the glycolytic flux was proven in the parental cell lines, while the inhibition in the resistant lines was minor.
The combined treatment Metf+PDT show a synergistic effect on cells viability, however the underlying mechanism appear to be contractidory: ASZ cells show a decrease of pAKT and p-340 p70S6K, indicating a suppression of the mTOR pathway. However, CSZ increase the expression of pAKT and 343 p-p70S6K, indicating an over activation of the mTOR pathway.
Other comments:
Fig. 1 A schematic diagram of the protcol followed to generated the 2 resistant cell lines woul help to understand the several passages to generate the cell lines. The authors should also explain the rationale of doing a subcutaneous implantation onto mouse model and explain why the ASZ line become less resistant once inoculated into the reciepint mouse. The results obtained indicated that 10GT-ASZ line was more sensitive to MAL-PDT than their corresponding P cells, and this is in contrast with the previous article. Why ?
Fig.1 Was the control performed by exposing the cells to the same intensity and duration of light withouth MAL ?
In the previous paper the authors show that 0.2 mM MAL for 5 h and subsequently exposed to variable doses of red light induced 15% cell survival, whereas in this study it does not seem to be the case. Only from 11.4 J/cm2 the same % of cell surival is reached. Can the authors comment on that ?
Table S1: please add the expected behaviour in a physiological human BCC to help the reader understand the characterization of the cell lines in comparison with a physiological environment, even better a comparison with a patient derived resitant BCC cell line...
Fig 3c: what are we look at? Both results and figure legend do not explain what this represenatitive picture is, authors should show the images of every group, even in the supporting materials.
The in vivo xenograph study was performed only on ASZ cell line. The synergistic effect of Met+PDT treatmetn seems to work in the paretnal cell line (and not in the resistant) but I’m really concerned on the very big standard deviation making the staitistical relevance hard to believe. Is the Metf+PDT a synergistic or addditive effect? In the reistant counterpart, the PDT drive an increase in tumor size, can the author comment more on that? The authors should add a deeper characterization of the explanted tumors with hystological analysis with cell death markers (e.g. Tunel assay) and statistical analysis.
Author Response
- Referee 2
The authors presented a very interesting study regarding the combined usage of PDT and the drug Metformin, tipically used to lower the blood sugar in type 2 diabetes. Metformin is in fact a first‐line antihyperglycaemic not commonly used to treat cancer, especially BCC tumor type, so this represent a novelty in the field.
In their previous paper the authors developed a very smart strategy to induce PDT resistant cell lines (“Characterisation of resistance mechanisms developed by basal cell carcinoma cells in response to repeated cycles of Photodynamic Therapy” Sci Rep. 2019; 9: 4835.) and in this second article they use their model to prove the possible combination of PDT and Metf to increase cell death in case of tumor resistance.
Nevertheless, in the study several hypotheses based on previous and related literature, have not been proven probably due to the nature of the murine cell lines selected, their variability and the lack of a positive control such as resistant patient derived tumor cells.
For instance, the authors evaluated the potential metabolic changes that occur in BCC cells resistant to PDT, associating the resistance with a glycolytic metabolism, in order to get a rationale for the usage of an antihyperglycaemic drug. The results show changes in the metabolism of resistant cells towards an aerobic glycolysis, neverthless although it has been described that the aerobic glycolytic metabolism is an advantage for the cancer cell proliferation, the authors found no differences in the proliferation rate between parental and PDT resistant cells.
In addition, the hypothesis that Metf works by inhibiting the glycolytic flux was proven in the parental cell lines, while the inhibition in the resistant lines was minor.
The combined treatment Metf+PDT show a synergistic effect on cells viability, however the underlying mechanism appear to be contractidory: ASZ cells show a decrease of pAKT and p-340 p70S6K, indicating a suppression of the mTOR pathway. However, CSZ increase the expression of pAKT and 343 p-p70S6K, indicating an over activation of the mTOR pathway.
We want to thank the reviewer for its comments on our MS.
Other comments:
- Fig. 1 A schematic diagram of the protocol followed to generated the 2 resistant cell lines woul help to understand the several passages to generate the cell lines. The authors should also explain the rationale of doing a subcutaneous implantation onto mouse model and explain why the ASZ line become less resistant once inoculated into the reciepint mouse. The results obtained indicated that 10GT-ASZ line was more sensitive to MAL-PDT than their corresponding P cells, and this is in contrast with the previous article. Why?
According to the suggestion made by the referee, we have added a supplementary figure S1 with a schematic diagram of the protocol followed to generate both resistant cell populations. We have modified the text to explain the methodology used for obtaining the tumor resistant cells (10GT) as follows (line: 300): “…population called 10GT [2], this process was carried under the basis that the in vivo microenvironment promotes the selection of cells with higher tumorigenic properties (da Silva-Diz et al. 2016. Cancer stem-like cells act via distinct signaling pathways in promoting late stages of malignant progression. Cancer research, 76(5): 1245-1259). Surprisingly, as indicated in the text (line 303), 10GT ASZ cells were more sensible to PDT that those inoculated into the mice. We do not know the reason for this unexpected behavior, which is now being investigated in our lab. In any case, we have decided to employ the most resistant cell populations: 10G for ASZ and 10GT CSZ.
- Fig.1 Was the control performed by exposing the cells to the same intensity and duration of light withouth MAL?
Yes, all the corresponding controls were performed for cell toxicity estimation. According to the comment indicated, we have included a sentence in the text to clarify this (line 90): “For all the experiments the corresponding controls were performed: untreated cells (cells without MAL or light irradiation) and cells treated with MAL (0.2 mM, 5 h) or red light alone (15.2 J/cm2); no cell toxicity was detected”.
- In the previous paper the authors show that 0.2 mM MAL for 5 h and subsequently exposed to variable doses of red light induced 15% cell survival, whereas in this study it does not seem to be the case. Only from 11.4 J/cm2 the same % of cell surival is reached. Can the authors comment on that ?
In the previous paper we selected 0.3 mM or 0.2 mM MAL for ASZ and CSZ cells treatment, respectively. In this work we have considered to use the same concentration (0.2 mM MAL) in both cases, being necessary to administrate higher red light doses for obtaining the same ASZ cell toxicity as in the previous paper. In the case of CSZ cells the differences observed in survival between the previous and the present work could be related with the culture confluence when PDT was applied; in the previous study PDT was applied when cells reached 50-60% of confluence and in the present work when cultures reached a 70% of confluence. As indicate by other authors, confluence of the cultures seems to be related with the response to the PDT treatments, as it happens in CSZ cells (see for instance: Delaey, E. et al. 1999. Confluence dependent resistance to photo-activated hypericin in HeLa cells. Int J Oncol. 14:759).
- Table S1: please add the expected behaviour in a physiological human BCC to help the reader understand the characterization of the cell lines in comparison with a physiological environment, even better a comparison with a patient derived resitant BCC cell line...
We thank the comment made by the reviewer, which is very opportune. In fact, we are working in studying the expression of the metabolic markers in human BCC in a translational study under the supervision of Dr Gilaberte and Dr González at the Miguel Servet Hospital (Zaragoza, Spain). In previous studies performed on BCC resistant to PDT, we have demonstrated that factors such as P53 and cyclin D1 expression are directly related with the response to PDT (Gracia-Cazaña et al. 2019. Biomarkers of basal cell carcinoma resistance to methyl-aminolevulinate photodynamic therapy. PLoS One. 24:14; Gracia-Cazaña et al. 2018. Photodynamic therapy: influence of clinical and procedure variables on treatment response in basal cell carcinoma and bowen disease. Acta Derm Venereol. 12:98). Since we do not have significant results on the expression of metabolic markers in biopsies of patients we prefer not performing a table with the data suggested by the reviewer.
- Fig 3c: what are we look at? Both results and figure legend do not explain what this represenatitive picture is, authors should show the images of every group, even in the supporting materials.
According to the suggestion made by the reviewer, we have modified the figure 3c, adding representative JC1 images for all the conditions evaluated (untreated and metformin treated cells).
- The in vivo xenograph study was performed only on ASZ cell line. The synergistic effect of Met+PDT treatmetn seems to work in the paretnal cell line (and not in the resistant) but I’m really concerned on the very big standard deviation making the staitistical relevance hard to believe. Is the Metf+PDT a synergistic or addditive effect? In the reistant counterpart, the PDT drive an increase in tumor size, can the author comment more on that? The authors should add a deeper characterization of the explanted tumors with hystological analysis with cell death markers (e.g. Tunel assay) and statistical analysis.
Since ASZ cell line responded better to the combined treatment, we decided to use only this cell line for the experiments in vivo, with mice. In vitro, the combined treatment showed a synergistic effect in parental and resistant populations of both cell lines. In mice, the combined treatment is more effective in tumors induced by inoculation of parental than those of resistant cells.
In xenografts induced by ASZ P, Metf significantly inhibited tumor growth, whereas the size of the tumors treated with PDT did not decrease. The combined treatment improved the efficacy observed after Metf application and the sizes of tumors were the smallest. In the xenografts induced by 10G, Metf did not significantly inhibit tumor growth, and the tumors treated with PDT showed a significant increase (as they are formed by cells resistant to PDT). The combined treatment induced a significant decrease in the size of the tumors.
We agree with the referee in relation with the error bars, they are quite large and perhaps this is the reason why we cannot indicate if the combined effect is additive or synergistic as we have done in vitro.
In relation with the characterization of the tumors, in a first attempt, we have changed the photographs of the H&E staining to better show the effects of the different treatments (figure 6c). We have also incorporated photographs showing positivity to Tunel assay of the sections obtaining from the untreated and treated tumors, clearly showing a higher positivity in tumors subjected to Metf + PDT, in P tumors. A deeper characterization of the tumors will be the objective of a next translational paper, which is being developed by testing the expression of different markers of interest.
Reviewer 3 Report
The manuscript titled ‘Metformin as an adjuvant to photodynamic therapy in resistant basal cell carcinoma cells’ by Mascaraque M., et al describes the effect of metformin, an anti-diabetus drug, on enhancing PDT effects on murine BCC cells in both in vitro and in vivo models. The study was carefully conducted and the results should be interesting to the Reader of Cancers. However, there are some points authors need to clarify before its publication.
General comments:
Topically Metvix-PDT of BCC is an established modality with a high complete response rate, but a tumor recurrence rate of about 20% has been seen after >12 months follow-up. Such recurrence may largely be due to the fact that Metvix can not penetrate enough into a whole nodular BCC lesion (even with prior curettage) and morpheaform of BCC. Light penetration is another issue for partial response. Tumor resistant clones may be developed after repeating PDT, but our experience does not show a high fraction of those recurrent cases. Combined PDT with a drug would improve the treatment effectiveness.
Specific comments:
Any special reason(s) to choose MAL instead of ALA in this study? Normally, ALA produces more PpIX than MAL in cells in both in vitro and in vivo models. During MAL incubation with cells, did DMEM contain 10% FBS. If yes, FBS can significantly reduce PpIX production. Did formaldehyde affect PpIX measurements in cells? Is there any reference for it? Perhaps provide the information on tumor size when PDT started. MAL was injected ip or iv? Was there any reason to give the light irradiation to tumor at 4 hrs after injection? It is true that 4-hr is for the ALA case. However, MAL peaks PpIX production in tissues in vivo at earlier time points (0.5 to 2 hrs) after administration. A poor quality of some immunoblots presented. In Figure 3b, perhaps more proper to present histograms instead of curves. In Figures 4b and 4b’ perhaps present a good example with isobolograms? Some typos.Author Response
- Referee 3
The manuscript titled ‘Metformin as an adjuvant to photodynamic therapy in resistant basal cell carcinoma cells’ by Mascaraque M., et al describes the effect of metformin, an anti-diabetus drug, on enhancing PDT effects on murine BCC cells in both in vitro and in vivo models. The study was carefully conducted and the results should be interesting to the Reader of Cancers. However, there are some points authors need to clarify before its publication.
General comments:
Topically Metvix-PDT of BCC is an established modality with a high complete response rate, but a tumor recurrence rate of about 20% has been seen after >12 months follow-up. Such recurrence may largely be due to the fact that Metvix can not penetrate enough into a whole nodular BCC lesion (even with prior curettage) and morpheaform of BCC. Light penetration is another issue for partial response. Tumor resistant clones may be developed after repeating PDT, but our experience does not show a high fraction of those recurrent cases. Combined PDT with a drug would improve the treatment effectiveness.
We thank the referee for the comments.
Specific comments:
- Any special reason(s) to choose MAL instead of ALA in this study? Normally, ALA produces more PpIX than MAL in cells in both in vitro and in vivo models.
MAL was the first approved prodrug for superficial and/or nodular BCC in Europe and in other countries (European Dermatology Forum Guidelines on topical photodynamic therapy. Eur J Dermatol. 2015. doi:10.1684/ejd.2015.2570) and its efficacy has been widely reported. In addition, there are studies showing no differences between ALA-PDT and MAL-PDT for BCC treatment in clinical practice (Kessels et al. 2018. Treatment of superficial basal cell carcinoma by topical photodynamic therapy with fractionated 5‐aminolaevulinic acid 20% vs. two‐stage topical methyl aminolaevulinate: results of a randomized controlled trial. British Journal of Dermatology, 178(5), 1056-1063). This is the main reason why we have selected MAL for the study.
- During MAL incubation with cells, did DMEM contain 10% FBS. If yes, FBS can significantly reduce PpIX production. Did formaldehyde affect PpIX measurements in cells? Is there any reference for it?
Cells were incubated with MAL in DMEM without FBS as indicated in line 386 of the M&M section: “For phototreatments, when cells reached a 60-70% confluence were incubated with 0.2 mM MAL in DMEM culture medium without FBS for 5 h.”
Previous experiments in our lab showed that fixation of HaCaT cells with formaldehyde in PBS for flow cytometry had no effects in PpIX measurements (Blazquez-Castro et al. 2012. Protoporphyrin IX-dependent photodynamic production of endogenous ROS stimulates cell proliferation. Eur. J. Cell. Biol. 91(3):216-223).
- Perhaps provide the information on tumor size when PDT started. MAL was injected ip or iv?
According to the suggestion made by the reviewer, we have included the information about tumor size when PDT was applied; MAL was administered by intratumoral injection to be sure that the compound is in the tumor (line 490): “The right tumor of each mouse was treated with PDT at day 15 after inoculation, when the tumor volumes reached a size of 100-150 mm3. For PDT, 50 µl PBS containing 2 mM MAL was administered by intratumoral injection …”.
- Was there any reason to give the light irradiation to tumor at 4 hrs after injection? It is true that 4-hr is for the ALA case. However, MAL peaks PpIX production in tissues in vivo at earlier time points (0.5 to 2 hrs) after administration.
We injected MAL into the tumors and we decided to wait 4 h prior to light irradiation to be sure that the cells of the tumor produced PpIX. In addition, other studies indicated an increased accumulation of PpIX at longer times than 3 h of incubation with MAL (see for instance: Leufflen, L. et al. 2018. Photodynamic diagnosis with methyl-5-aminolevulinate in squamous intraepithelial lesions of the vulva: Experimental research. PloS one, 13(5)).
- A poor quality of some immunoblots presented. In Figure 3b, perhaps more proper to present histograms instead of curves.
We have improved the quality of immunoblots. Regarding figure 3b, we are showing the results in histograms as suggested.
- In Figures 4b and 4b’ perhaps present a good example with isobolograms? Some typos.
We really want to thank the suggestion made by the reviewer, but we consider that the curves graphs shown in the figure 4b and 4b´are clear enough to demonstrate the synergistic effect of Metf and PDT.
Reviewer 4 Report
Marta Mascaraque and co-authors investigated the ability of the antidiabetic type II compound metformin to sensitize resistant cell models of Basal Cell Carcinoma (BCC) to Photodynamic Therapy (PDT). The aim of the study is to propose metformin as an adjuvant to photodynamic therapy with methyl-aminolevulinate (MAL-PDT) in resistant BCC. In this study, the mouse BCC ASZ and CSZ cells were used as sensitive (parental) cells and to obtain resistant cell BCC model cells 10G ASZ and 10GT CSZ. The authors decided to treat these cells with metformin based on the evidence of a different energy metabolism between sensitive and resistant BCC cell lines that could influence the response to therapy. The resistant carcinoma cells prefer glycolysis instead mitochondrial respiration to produce energy. The metabolic effects of metformin are not the same in all cell lines. In P ASZ metformin induced an increase of OCR in contrast to the decrease obtained in 10G ASZ, P CSZ.
The adjuvant effect of was also evaluated in vivo in a xenograft animal model. In this case metformin contributes to the inhibition of tumor growth.
The propose of metformin as adjuvant to photodynamic therapy resistant BCC is very interesting since metformin is a well known and approved drug. However, this research is not well performed. The results are not clearly described, and many figures have a poor quality. In my opinion, the role of metformin as adjuvant in BCC treatment, is not well highlighted.
For all these reasons, I suggest accepting the manuscript after major revision
In details:
- Figure 2, panel b and c, show the western blot of β-F1 ATPase, GAPDH and PKM2 protein levels in ASZ and CSZ sensitive and resistant cells. The level of α tubulin is used as loading control. The western blot image of α tubulin is the same in figure 2b and figure 2c. Why? Due to the similar MW of β-F1 ATPase and PKM2, these 2 proteins must be analysed in 2 different SDS PAGE each of them with its loading control.
- Figure 2b. the quality of β-F1 ATPase WB image is very poor. The shape of the immunoblot bands is irregular, and it can hardly use to quantification. I suggest performing new western blot.
- Line 140. The statement is incorrect since is in contrast with data presented in figure 2. The glycolytic metabolism in higher only in resistant ASZ cell line. (see also Table S1)
- Line 153. The red / green fluorescence ratio was significantly increased in 10GT of CSZ cells after 24h of Metf treatment, indicating that Metf causes hyperpolarization of mitochondrial membrane in resistant cells by inhibition of the respiratory chain “. It is incorrect. The inhibition of the respiratory chain leads to mitochondrial membrane depolarization due to the block of electron transport.
- Line 160. “However, in 10GT of CSZ, the β‐F1‐ATPase/GAPDH ratio was significantly increased after Metf treatment, suggesting a higher OXPHOS.” This comment in in contrast with the sentence reported in Line 153.
- Line 165. “The OCR after Metf treatment showed a significant increase in the maxima respiration (after addition of B, DNP) in ASZ P cells…”. In figure 3g the OCR of metformin treated ASZ P cells is similar to control.
- Figure 3c. The representative JC1 images for all cell lines must be showed.
- Figure 3d. The level of PKM2 in 10G ASZ should be statistically higher than in P ASZ as reported in Figure 2d.
- Figure 3e. The quality of western blot images is poor (PKM2 and α tubulin). Moreover, I wonder if all these images are obtained from the same SDS PAGE experiment. As stated above, due to similar MW of β-F1 ATPase and PKM2, these 2 proteins must be analysed in 2 different SDS PAGE each of them with its loading control.
- The level of PKM2 in P CSZ should be higher than 10GT CSZ as reported in figure 2c.
- Line 216. “All populations showed higher intracellular PpIX content after 5 h of MAL incubation with respect to untreated cells, being significantly higher in both CSZ….” I do not agree. The PpIX content is statistically significant increased only in CSZ. No effect of MAL treatment is evident in ASZ cells (figure 5a).
- Figure 5b. The western blot of total AKT and total p70S6K must be reported. The level of p-AKT (and p-p70S6K) is the ratio p-AKT/AKT.
- Figure 5b. Concerning the western blot on CSZ cells, the quality of p-AMPK and AMPK is very poor, and it renders the quantification analysis very difficult. It should be improved.
- Figure 5b. The histograms of p-AKT and p-p70S6K western blot are missing.
- Line 329. “However, Metf did not increase the accumulation of PpIX into cells, so this is not the mechanism by which the combination of these treatments increases its cytotoxicity.” The author should discuss deeply these data suggesting hypotheses.
Author Response
- Referee 4:
Marta Mascaraque and co-authors investigated the ability of the antidiabetic type II compound metformin to sensitize resistant cell models of Basal Cell Carcinoma (BCC) to Photodynamic Therapy (PDT). The aim of the study is to propose metformin as an adjuvant to photodynamic therapy with methyl-aminolevulinate (MAL-PDT) in resistant BCC. In this study, the mouse BCC ASZ and CSZ cells were used as sensitive (parental) cells and to obtain resistant cell BCC model cells 10G ASZ and 10GT CSZ. The authors decided to treat these cells with metformin based on the evidence of a different energy metabolism between sensitive and resistant BCC cell lines that could influence the response to therapy. The resistant carcinoma cells prefer glycolysis instead mitochondrial respiration to produce energy. The metabolic effects of metformin are not the same in all cell lines. In P ASZ metformin induced an increase of OCR in contrast to the decrease obtained in 10G ASZ, P CSZ.
The adjuvant effect of was also evaluated in vivo in a xenograft animal model. In this case metformin contributes to the inhibition of tumor growth.
The propose of metformin as adjuvant to photodynamic therapy resistant BCC is very interesting since metformin is a well known and approved drug. However, this research is not well performed. The results are not clearly described, and many figures have a poor quality. In my opinion, the role of metformin as adjuvant in BCC treatment, is not well highlighted.
For all these reasons, I suggest accepting the manuscript after major revision.
We want to thank the comments made and we have been trying to solve the objections indicated as described in the next points.
In details:
- Figure 2, panel b and c, show the western blot of β-F1 ATPase, GAPDH and PKM2 protein levels in ASZ and CSZ sensitive and resistant cells. The level of α tubulin is used as loading control. The western blot image of α tubulin is the same in figure 2b and figure 2c. Why? Due to the similar MW of β-F1 ATPase and PKM2, these 2 proteins must be analysed in 2 different SDS PAGE each of them with its loading control.
The reviewer is right and α-tubulin loading control has been included for each western blot as pointed out. The expression of each marker has been analyzed to that of its loading control. In the case of β-F1 ATPase and PKM2, they both were analyzed in 2 different SDS western with its loading control.
- Figure 2b. the quality of β-F1 ATPase WB image is very poor. The shape of the immunoblot bands is irregular, and it can hardly use to quantification. I suggest performing new western blot.
According to the suggestion made by the reviewer, we have included new immunoblots to better show the expression of β-F1 ATPase in the figure 2b.
- Line 140. The statement is incorrect since is in contrast with data presented in figure 2. The glycolytic metabolism in higher only in resistant ASZ cell line. (see also Table S1)
The reviewer is fully right. Resistant CSZ did not show a higher glycolysis than parental cells (since PKM2 and lactate production are similar between both populations), only a lower OXPHOS has been observed. We have modified the text as follows (line 146): “The results shown above indicated that in the PDT resistant cells, the OXPHOS metabolism was lower than that of parental cells. In addition, resistant ASZ population showed a higher glucose metabolism than that of parental cells.”
- Line 153. The red / green fluorescence ratio was significantly increased in 10GT of CSZ cells after 24h of Metf treatment, indicating that Metf causes hyperpolarization of mitochondrial membrane in resistant cells by inhibition of the respiratory chain “. It is incorrect. The inhibition of the respiratory chain leads to mitochondrial membrane depolarization due to the block of electron transport.
We do not completely agree with the reviewer’s comments. It is true that the inhibition of the respiratory chain could induce mitochondrial depolarization provided that, no electrons are being funneled through Complex II and the other dehydrogenases that participate in the respiratory chain. However, the increase of the ref/green fluorescence ratio indicates hyperpolarization. Metformin is an inhibitor of Complex I of the electron transport chain and there are several references that correlate the inhibition of this complex with mitochondrial hyperpolarization ((1)Forkink, M. et al. (2014). Mitochondrial hyperpolarization during chronic complex I inhibition is sustained by low activity of complex II, III, IV and V. (BBA)-Bioenergetics, 1837(8), 1247-1256; (2) Bonsi, P. et al. (2004). Early ionic and membrane potential changes caused by the pesticide rotenone in striatal cholinergic interneurons. Experimental neurology, 185(1), 169-181; (3) Xin, G et al. (2016). Metformin uniquely prevents thrombosis by inhibiting platelet activation and mtDNA release. Scientific reports, 6, 36222.).
- Line 160. “However, in 10GT of CSZ, the β‐F1‐ATPase/GAPDH ratio was significantly increased after Metf treatment, suggesting a higher OXPHOS.” This comment in in contrast with the sentence reported in Line 153.
We fully agree with the reviewers comment. We have modified line 169 accordingly, for better understanding: “[…] suggesting a higher ATP production”. This result was in agreement with the increase in the mitochondrial membrane potential observed with JC-1 staining in 10GT of CSZ cells treated with Metf.
- Line 165. “The OCR after Metf treatment showed a significant increase in the maxima respiration (after addition of DNP, B) in ASZ P cells…”. In figure 3g the OCR of metformin treated ASZ P cells is similar to control.
The sentence mentioned by the reviewer is related with the “maximal respiration rate” which is shown in figure 3f after addition of the uncoupler DNP. DNP mimics a physiological “energy demand” by stimulating the respiratory chain at maximum capacity, inducing a rapid oxidation of substrates (sugars, fats, amino acids) to obtain this metabolic challenge.
The figure 3g shows the oligomycin sensitive respiration (OSR), which represents the activity of synthesis of ATP by complex V, coupled to the respiratory chain. OSR is related with the ATP produced by the mitochondria contributing to obtain the energy needed by the cell.
Therefore, maximal respiration rate and OSR are two different parameters.
- Figure 3c. The representative JC1 images for all cell lines must be showed.
According to the suggestion made by the reviewer, we have modified the figure 3c, adding representative JC1 images for all the conditions evaluated (untreated and metformin treated cells).
- Figure 3d. The level of PKM2 in 10G ASZ should be statistically higher than in P ASZ as reported in Figure 2d.
We thank the reviewer for noticing these mistakes. Yes, the PKM2 expression in ASZ 10G is statistically higher than in P ASZ which is now properly corrected in figure 3d; the significant difference is now shown by an asterisk.
- Figure 3e. The quality of western blot images is poor (PKM2 and α tubulin). Moreover, I wonder if all these images are obtained from the same SDS PAGE experiment. As stated above, due to similar MW of β-F1 ATPase and PKM2, these 2 proteins must be analysed in 2 different SDS PAGE each of them with its loading control.
According to the suggestion made by reviewer, we have changed the western blots images to improve its quality. In the same way as in the comment number 1, β-F1 ATPase and PKM2, were analyzed in 2 different SDS western with its loading control.
- The level of PKM2 in P CSZ should be higher than 10GT CSZ as reported in figure 2c.
We thank the reviewer for noticing this new mistake which has been now corrected. We have checked our results and corrected the error bars; in both figures the value of PKM2 in 10GT CSZ was diminished.
- Line 216. “All populations showed higher intracellular PpIX content after 5 h of MAL incubation with respect to untreated cells, being significantly higher in both CSZ….” I do not agree. The PpIX content is statistically significant increased only in CSZ. No effect of MAL treatment is evident in ASZ cells (figure 5a).
Thank you for this comment. The production of PpIX after MAL incubation in ASZ cells (P and 10G) seemed to be evident, although this was not statistically significant with respect to untreated cells. In agreement with your comment, we have changed the sentence in the text as follows (line 224): “All CSZ populations showed higher intracellular PpIX content after 5 h of MAL incubation with respect to untreated cells, being statistically significant. In the case of ASZ cells (P and 10G), the production of PpIX, after MAL incubation, seemed to be evident, although this was not statistically significant with respect to untreated cells.”
- Figure 5b. The western blot of total AKT and total p70S6K must be reported. The level of p-AKT (and p-p70S6K) is the ratio p-AKT/AKT.
Western blots of total AKT and p70S6K have been incorporated to the figure 5b as indicated.
- Figure 5b. Concerning the western blot on CSZ cells, the quality of p-AMPK and AMPK is very poor, and it renders the quantification analysis very difficult. It should be improved.
We have modified the western blots as the reviewer suggested to improve its quality.
- Figure 5b. The histograms of p-AKT and p-p70S6K western blot are missing.
Since we have incorporated western blot for total AKT and p70S6K, we have also performed the corresponding histograms showing the ratio between phosphorylate and the total protein expression. This has been incorporated in the figure 5b as suggested.
- Line 329. “However, Metf did not increase the accumulation of PpIX into cells, so this is not the mechanism by which the combination of these treatments increases its cytotoxicity.” The author should discuss deeply these data suggesting hypotheses.
Yes, the reviewer is all right and this is a very interesting subject to be studied. Most of the published papers link PpIX production with the response to PDT. This was not observed in our experiments and Metf did not increase the accumulation of PpIX in BCC cells. Therefore, this is not the mechanism by which the combinations of these treatments increase the cytotoxicity. Most probably the modulation of the mTOR pathway and OXPHOS by Metf (which also occurs in mitochondria were PpIX is produced), is implicated. This sentence has been added in the Discussion section (line 366). In any case, this aspects is word while to analyze separately.
Round 2
Reviewer 2 Report
I would like to acknowledge to the authors, I'm satisfied with the answers to my concerns and issues.
Reviewer 4 Report
Marta Mascaraque and co-authors have deeply revised the manuscript entitled “Metformin as an adjuvant to photodynamic therapy in resistant basal cell carcinoma cells” addressing all the reviewer's comments. This revised manuscript is suitable for the publication in Cancers journal.